# Importance of occupation in the increase of gender differences in the advance of telework in Spain

**Pilar Ortiz García**[ID]°, **Salvador Manzanera-Román**[ID]°*

Department of Sociology, University of Murcia, Murcia, Spain

° These authors contributed equally to this work.
* smanzanera@um.es

## Abstract

The COVID-19 pandemic significantly altered forms of labor. In this analysis, which focuses on the Spanish labor market, we consider telework from a gender-based perspective in order to identify whether there is a gender gap in this work form and, if so, whether this is impacted by occupation. The methodology is quantitative in nature, and we apply logistic regression based on data from Eurofound's European Working Conditions Survey. We investigate how factors including labor characteristics, education, and life stage and cohabitation affect the propensity to engage in telework. The findings reveal the dominance of women in telework, moderated by labor characteristics (occupation, type of contract, and working day) and educational variables.

## Introduction

Technology has significantly propelled the digitalization process of modern society [1,2]. In doing so, it has created a new social model with specific forms of structuration, communication, and socialization [3,4], both in traditional, analog spaces and new, virtual ones.

The COVID-19 pandemic triggered a specific digital shift [5]. Compared to earlier digital shifts, this transformation entailed specific characteristics, such as speed, intensity, and a high capacity to impact daily habits and practices, all of which led to the substantial alteration of previous social structures and forms [6,7]. During the pandemic, digital technologies became essential for maintaining daily life, especially during lockdowns [8]. Many saw increased importance in digital technologies, with 56.4% of Spaniards believing that that these technologies would become even more central in their lives once the health crisis was over and normality was restored [9]. In the European Union (EU), up to 81.0% of individuals surveyed stated that digital technologies would be an important part of their lives by 2030 [10].

In this context, telework, which became more prevalent throughout all EU countries as a consequence of the pandemic, has created significant opportunities

**Data availability statement:** The data underlying the results presented in the study are available from Eurofound - European Working Conditions Telephone Survey database, 2021 (https://doi.org/10.5255/UKDA-SN-9026-3).

**Funding:** The author(s) received no specific funding for this work.

**Competing interests:** The authors have declared that no competing interests exist.

to improve working conditions. However, telework can also consolidate existing practices that perpetuate gender inequality in the labor market. Several studies have eschewed technological determinism and relativized the power of new technologies, since they impose changes that maintain the respective positions of the privileged and the disadvantaged classes in the occupational structure of the labor market [11].

Telework and its implications can be studied according to three complementary axes of analysis. The first considers telework as a productive strategy, and one of the many forms of the productive flexibility paradigm resulting from the Fordist regulation model [12]. The second assesses labor relations and conditions related to telework, including legal and regulatory aspects and those pertaining to training and promotion [13]. The third axis, which entails a psychosocial perspective, incorporates the personal consequences of telework for employees.

Although several studies have analyzed the evolution of telework and the factors moderating its growth [14,15], few have taken gender into account using a transversal perspective. The originality of the present paper lies in the fact that it reveals gender-related similarities and differences in the progression of telework, taking into account moderating factors via sociodemographic variables, such as education, life stage and cohabitation, and labor characteristics (occupation, contract type, and working arrangement). This analysis helps to elucidate labor-related issues in relation to the work–life balance-related issues traditionally associated with the gender-based approach.

The remainder of the article proceeds as follows. Next, we explain our theoretical approach. We then analyze the evolution of telework in Spain compared to that of the EU at large. Subsequently, we explain our methodology, and finally, we discuss our study's results, and provide conclusions.

## 1. Telework from a gender perspective

According to the abovementioned three axes of analysis, gender is a transversal element that can be used to study not only the conditions of this work modality but also its consequences. In the context of the COVID-19 pandemic, which impacted multiple areas of life, considering gender is particularly relevant considering that crises may increase inequality in the labor market to the detriment of underrepresented social groups, such as women [16–21].

According to the first axis of analysis, telework is a flexible production strategy. Studies have suggested that telework has both positive and negative consequences. Positive outcomes of flexible production systems include increased opportunities to adapt to the worker's specific needs by, for instance, cutting commuting time and increasing autonomy and work–family balance [20,22–28]. Negative aspects include the fact that telework, despite being legally regulated, suffers from certain regulatory gaps due to the fact that it is commonly performed from home [17,29,30].

In terms of jobs that can be performed remotely, studies have concluded that differences in the feasibility of telework depend primarily on the characteristics of each occupation [31,32], as factors such as the digitization of tasks, the use

of specialized tools, the need for face-to-face interaction, and security requirements influence the viability of this work form. For example, knowledge-based occupations, such as programming or accounting, can easily be adapted to remote work, while those requiring physical presence, such as manufacturing or face-to-face healthcare, present greater limitations. In addition, autonomy in work management and the need for supervision play key roles in the effectiveness of telework.

Concerning employment relationships, which belong to the second axis of analysis, research on the effects of telework has detected risks such as a decrease in the visibility of, and appreciation for, the work performed [33]. There is also a risk of increased gaps between representation and decision-making power in firms, which may exacerbate the already serious problem of the glass ceiling [34–36], and reduce promotion possibilities [37–40]. Relatedly, education level is a key factor in how individuals adapt to telework because it influences their comfort using new technologies, autonomy in time management, and ability to solve problems independently. Specifically, according to Kley and Reimer (2023), workers with higher levels of education tend to perform better in telework environments because they possess advanced digital skills and greater organizational capacity, which allow them to maintain high levels of productivity without direct supervision [41]. In addition, education fosters cognitive flexibility, facilitating the transition to telecommuting-based work models [42].

Positive aspects of telework regarding the second axis of analysis, which deals with working conditions and labor relations, are also pertinent. Research has suggested that telework reduces work stress, enhances creativity, boosts independence and autonomy, makes working hours more flexible, and enhances job satisfaction due to improvements in areas such as time management (e.g., due to the reduced or eliminated need to commute, though some studies have indicated that can be a negative "rebound" effect since commuting distances become longer; [43]). Another positive aspect is the improved work–life balance, which can in turn increase productivity and reduce absenteeism [27,44–47]. In the same vein, research has analyzed telework's potential to counter the job losses caused by the COVID-19 pandemic. However, evidence shows that this effect has been more significant for men than for women [48].

On the psychosocial consequences, which lie on the third axis of analysis, research has examined issues related to the difficulties posed by overexploitationor, more specifically, self-exploitation and digital disconnection. One of the most prominent risks is technostress arising from work time encroaching on personal time (creating a blurring effect). Although these issues are common to both men and women, they become more problematic for women, who are expected to carry out a greater number of diverse tasks when working from home and also carrying out family duties [49,50]. Haddon and Brynin (2005) indicated that this work modality is not just another factor but allows us to divide labor practices according to the categories corresponding to the fields of gender, education, occupation, and wage by including the gaps related to those fields in such categories. Another risk of telework is isolation [51]. Again, while this risk affects both men and women, it is more pronounced for women, as gender roles place much of the child-rearing work on women, reducing the amount of time they can spend cultivating social relations. The extent to which isolation is an issue also depends on the establishment of boundaries between work and life, as such boundaries can help to mitigate conflict between the two spheres [52].

The effects of telework depend on aspects such as willingness to adopt this form of employment [53], perceptions of its positive consequences, and the corporate labor policy, which, together with the flexibilization, must incorporate measures to protect employees, such as safety, promotion, and gender equality measures [38,53,54]. Research has shown that there is no such willingness in the case of women, [55], but rather the acceptance of socially attributed obligations in the face of the slow progress of domestic work co-responsibility.

Overall, the propensity to engage in telework, which is considered here as a function of gender, is conditioned by the multiple variables included in the three axes of analysis (structural, industrial relations, and psychosocial). Here, we seek to elucidate how these variables moderate the propensity to engage in telework given the increase in telework among women in the EU, and in Spain in particular.

## 1.1. The evolution of telework in the European Union

Analyses of the evolution of telework—understood here as working from home as a typical, rather than occasional, arrangement—in terms of gender in the EU (27) indicates that more women than men engage in telework. In the wake of the COVID-19 pandemic, these differences have increased, reaching 1.8 and 1.9 percentage points in 2020 and 2021 respectively (Table 1). Observation of the evolution of the gender gap in telework across EU member states indicates three types of behavior patterns:

- More women than men engage in regular telework, with a growing gender gap in the post-pandemic period, especially in countries such as France, Belgium, Portugal, Estonia, Greece, the Czech Republic, and Poland.

- Regular telework pattern that has been feminized after the pandemic, where before the pandemic more men than women would engage in telework. This has been the case in countries with a scissor-shaped gender gap, including Ireland, Sweden, Finland, the Netherlands, Italy, Lithuania, and Spain. The case of Spain is elaborated in Table 2.

**Table 1. Evolution of the gender gap in telework in the EU Member States (2012–2021).**

| | Number of Women (%) | | | | Gender Difference | | | |
|---|---|---|---|---|---|---|---|---|
| | 2012 | 2016 | 2020 | 2021↓ | 2012 | 2016 | 2020 | 2021 |
| Ireland | 3.6 | 3.0 | 21.7 | 32.5 | 2.3 | 0.7 | -0.4 | -1.0 |
| Luxembourg | 12.6 | 13.7 | 23.9 | 28.4 | -2.2 | -3.1 | -1.4 | -0.5 |
| Sweden | 4.1 | 4.7 | 6.8 | 28.0 | 0.8 | 0.7 | -0.8 | -1.9 |
| Belgium | 9.1 | 7.3 | 18.4 | 27.7 | 0.3 | -0.2 | -2.2 | -2.8 |
| Finland | 8.5 | 10.4 | 25.5 | 25.2 | 2.6 | 3.0 | -0.7 | -0.8 |
| Netherlands | 9.6 | 11.7 | 16.7 | 22.7 | 3.6 | 3.3 | 2.1 | -0.4 |
| Denmark | 10.6 | 7.5 | 17.0 | 18.9 | 2.1 | 1.7 | 0.0 | -1.5 |
| France | 13.5 | 8.5 | 17.6 | 18.7 | -3.7 | -3.2 | -3.6 | -3.4 |
| Malta | 2.2 | 4.8 | 18.2 | 18.6 | -0.6 | -2.0 | -5.8 | -6.3 |
| Estonia | 5.6 | 6.3 | 13.6 | 17.1 | 0.2 | -0.9 | -2.7 | -4.2 |
| Germany | 3.8 | 3.4 | 13.2 | 16.8 | -0.5 | -0.4 | 0.7 | 0.4 |
| Austria | 10.9 | 10.8 | 18.9 | 16.7 | -1.2 | -1.7 | -1.4 | -1.6 |
| Portugal | 7.0 | 6.7 | 14.7 | 15.7 | -1.4 | -0.7 | -1.5 | -2.4 |
| EU (27) | 5.9 | 5.1 | 13.0 | 14.4 | -0.7 | -0.6 | -1.8 | -1.9 |
| Latvia | 1.9 | 2.7 | 4.2 | 12.5 | 0.1 | -0.3 | 0.5 | -3.1 |
| Slovenia | 8.1 | 8.7 | 8.9 | 11.9 | -2.8 | -2.3 | -2.7 | -2.5 |
| Lithuania | 3.8 | 2.5 | 6.3 | 10.4 | 0.5 | 0.4 | -1.8 | -2.4 |
| Spain | 4.4 | 3.4 | 12.1 | 10.0 | 0.0 | 0.2 | -2.2 | -0.9 |
| Italy | 2.9 | 3.1 | 14.3 | 9.9 | 0.7 | 0.4 | -3.6 | -2.8 |
| Greece | 3.0 | 3.1 | 9.1 | 9.4 | -1.5 | -0.9 | -3.6 | -4.7 |
| Czechia | 4.3 | 4.5 | 7.9 | 8.6 | -1.6 | -1.2 | -1.2 | -2.6 |
| Poland | 5.0 | 5.5 | 10.5 | 7.8 | -0.8 | -0.3 | -2.9 | -1.6 |
| Cyprus | 1.2 | 1.8 | 5.3 | 7.3 | -0.3 | -0.3 | -1.5 | -1.1 |
| Slovakia | 4.0 | 3.7 | 6.8 | 7.0 | -0.9 | -0.8 | -2.1 | -0.7 |
| Croatia | 1.1 | 1.7 | 4.2 | 5.8 | -0.3 | -0.6 | -2.2 | -2.1 |
| Hungary | 3.3 | 3.1 | 4.2 | 5.0 | -0.4 | -0.3 | -1.1 | -0.9 |
| Bulgaria | 0.6 | : | 1.9 | 3.4 | -0.1 | | -1.3 | -1.2 |
| Romania | 0.6 | 0.6 | 3.7 | 3.3 | -0.3 | -0.2 | -2.0 | -1.6 |

*Note*: Values are derived by subtracting the percentage of men from that of women who telework. A positive value indicates that the proportion of men is higher than that of women, while a negative value indicates that the proportion of women is higher than that of men.

*Source*: Own elaboration based on data from Labour Force Survey, Eurostat (2021) [57].

**Table 2. Evolution of the gender gap in telework in Spain (2012–2021).**

| | 2012 | 2013 | 2014 | 2015 | 2016 | 2017 | 2018 | 2019 | 2020 | 2021 |
|---|---|---|---|---|---|---|---|---|---|---|
| Number of women (%) | 4.4 | 4.5 | 4.3 | 3.5 | 3.4 | 4.2 | 4.2 | 4.9 | 12.1 | 10.0 |
| Gender difference | 0.0 | -0.4 | -0.1 | 0.1 | 0.2 | 0.2 | 0.1 | 0.0 | -2.2 | -1.0 |

*Source*: Own elaboration based on data from Labour Force Survey, Eurostat (2021) [57].

- Intense telework pattern that has been masculinized after the pandemic. The evolution of the gender gap in the form of inverse scissors occurs only in Germany. This country also differs from other EU countries in regulatory matters [56].

Despite the existence of the abovementioned patterns, in the EU member states there has been a trend toward more women than men engaging in telework.

As shown in Table 2, in the case of Spain there has been an increase in women engaging in regular telework since 2020, which aligns with the countries of the second pattern described above. Prior to this, the proportion of men and women engaging in telework was comparable, although in most years there were slightly more men than women. Subsequent to the confinement induced by the pandemic, the gap between men and women widened significantly: While the pre-pandemic period exhibited no significant disparity between men and women, the year 2020 marked an increase of 2.2 percentage points, and 1 percentage point by the year 2021. Therefore, the case of Spain is distinctive compared to other countries within the same pattern. The substantial increase in telework among women that arose due to the pandemic was further fueled by the unique characteristics of the Spanish labor market.

In this context, during the COVID-19 pandemic, as highlighted above, there was a shift from limited to more intensive use of telework in order to maintain economic activity [14,58]. At the time of the telework boom in Spain, Spanish workers perceived telework as inevitable; although only one third of the Spanish workforce could engage in telework due to the nature of their job [59], 28.8% thought that the number of jobs that would be performed by means of telework would increase [9].

The profile of teleworkers in the EU, and Spain particularly, during the COVID-19 pandemic was broadly maintained with respect to the previous period [60]. Specifically, teleworkers were highly educated white-collar workers, working full time with a permanent contract, mainly in the service sector [61,62]. However, a low level of innovation has been observed in the organization of telework, with practices typical of the analog environment being maintained in the virtual environment [63], although the importance of flexible labor has been highlighted after the approval of Law 10/2021 [64] on telework [65].

Taking into account the particularity of the Spanish case and the existing literature on telework from a gender perspective, this research adopts a cross-sectional approach. Using data from the European Working Conditions Survey, we seek to validate hypotheses on the moderating effect of the propensity to telework as a function of variables such as education, life stage (family structure and cohabitation) and work history (occupation, type of contract and working hours).

This study provides a novel perspective by analyzing how these factors influence the adoption of telework within the specific structure of the Spanish labor market. In doing so, it fills a gap in the literature by addressing insufficiently explored dimensions. In addition, its findings may contribute to the development of new theories on telework, as well as to the identification of areas that require further analysis. Finally, the results of this research can serve as a basis for the design of more effective policies that promote better working conditions, thereby fostering improved social welfare.

## 2. Materials and methods

Our empirical analysis utilizes a quantitative methodology based on data extracted from the special edition of the European Working Conditions Survey (EWCS) published by Eurofound (2021) [66]. The fieldwork was conducted from March

to November 2021, thus marking one year after the initial onset of the confinement measures implemented in response to the pandemic. Consequently, the repercussions of the pandemic on telework practices were recorded in this edition of the EWCS. The total sample size amounts to 71,764 working people, 2,903 of which were located in Spain. As the result of a refinement process to ensure consistency with the objectives of the study, such that only relevant variables and cases to our analysis were included, we obtained a final sample of 1,320 individuals.

This reduction was partly due to the limitations inherent in the EWCS dataset, such as the multi-level structure of responses, which in some cases did not align directly with the variables in the model. In order to address this issue and prevent systematic errors or biases resulting from ambiguous response assignment, cases that could not be accurately categorized were excluded. This approach was adopted to ensure the validity of the results obtained by avoiding the artificial inflation of the sample size.

In order to address the issue of missing data, a structured approach was adopted. Initially, an assessment was conducted to determine the extent and pattern of missing values in key variables. Cases with missing values in the dependent variable (teleworking status) were excluded, as they could not contribute to the analysis. For the independent variables, a complete-case analysis (listwise deletion) was applied, with observations retained only if complete data was available for all predictors. The selection of this method was driven by the necessity to preserve both the consistency and interpretability of the model, whilst ensuring that the final sample accurately reflected the population under study.

A sensitivity check was conducted to compare key characteristics (such as gender, occupation, and education level) between the full Spanish sample (2,903) and the final analytical sample (1,320). The distributions remained largely consistent, suggesting that the exclusion of incomplete cases did not introduce substantial bias.

**2.1. Approach and hypotheses.** This research analyzes individuals' propensity to engage in telework based on several predictive variables. It also seeks to determine whether there are significant gender-based differences in this regard. Fig 1 illustrates the research approach.

According to the model and the literature review, we propose several hypotheses.

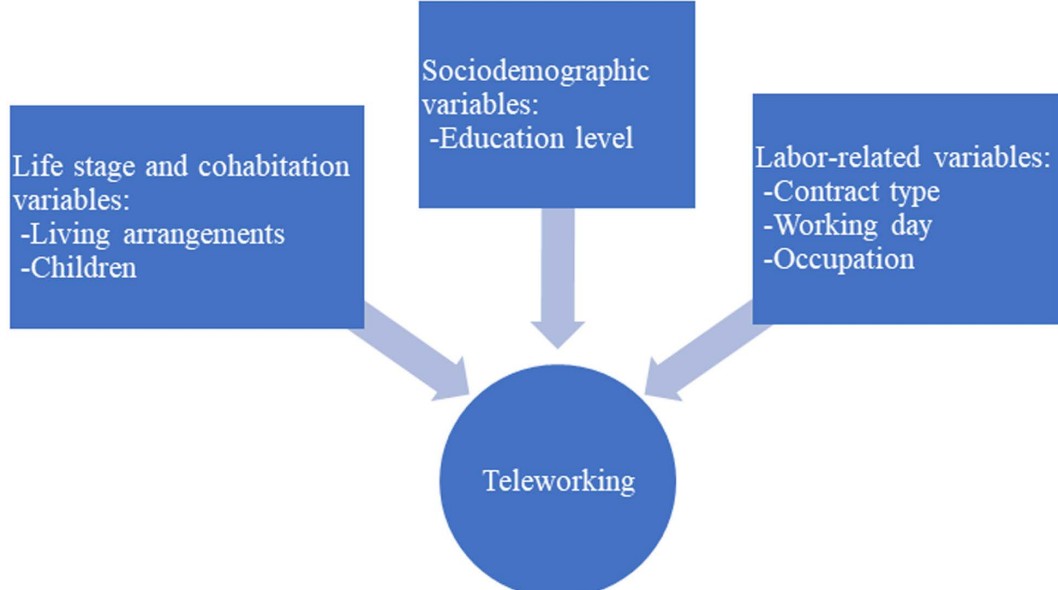

**Fig 1. Summary of the approach to analyze telework according to the variable type.** *Source*: Own elaboration.

*H1: Propensity to engage in telework is moderated by household composition (presence or absence of children, cohabitation vs. living alone).*

In households with children, the responsibilities for care and housework tend to fall on women, which increases their propensity to engage in telework as a strategy to reconcile these demands [67,68]. On the other hand, in households without children, gender differences in the propensity to engage in telework tend to be less pronounced. Furthermore, in households with two or more people, assigned roles may amplify or reduce these differences [24,69–72].

*H2: The occupational category, defined by level of responsibility and skill, moderates gender differences in the propensity to engage in telework.*

Women in higher-skilled and higher-responsibility occupations are more likely to engage in telework due to the flexibility this work form offers and the perception of trust involved [41,53,62,73,74]. In contrast, in low-skilled occupations or occupations with lower levels of autonomy, gender differences in the propensity to engage in telework are more marked, as these positions usually offer fewer options for work flexibility [75,76].

*H3. Educational level moderates the propensity to engage in telework.*

People with higher education are more likely to engage in telework, given that roles held by more highly educated individuals tend to be less dependent on face-to-face work and more compatible with technological tools [41,72,77,78]. In contrast, people with lower levels of education tend to occupy jobs that require manual skills or routine activities, limiting their ability to work remotely.

## 2.2. Description of variables and analysis method

Telework serves as a dependent variable in the analysis model. Although the Eurofound survey does not ask directly about telework, it does include the following question (QM35E): "How often have you worked in any of the following locations during the last 12 months?" The question includes five responses options ("never," "rarely," "sometimes," "often," and "always"). We determined the telework variable, which is considered as a nominal variable, based on three factors (ICT should be used either "always" or "often"; employees' job must be teleworkable; and employees should work from home either "rarely," "sometimes," "often," or "always"). The resulting variable was then recoded with the following values: full-time telework (employees always work from home), partial telework (employees often work from home, or often work from home and work from their employer's premises at least sometimes), occasional telework (employees sometimes or rarely work from home), or employer's premises with some degree of teleworkability (employees always work from their employer's premises but their job can be done remotely).

The following independent variables were considered:

- *Gender* (Q2new): nominal variable with values equal to "man" or "woman."

- *Living arrangements* (cohabitation): nominal variable. According to Eurofound (2021), this variable includes the following values: "single without children," "single with children," "two adults without children," "two adults with children," "more than two adults without children," and "more than two adults with children." We recoded this variable to obtain the values "living alone" and "living with partner and others" because there is a strong correlation between these and the children variable, according to the fact that it has identical values as their coding shows.

- *Children* (calculated by Eurofound, 2021 [66]): nominal variable with values equal to "no children" or "one or more children."

- *Education level* (Q106): nominal variable with values equal to "early childhood education," "primary education," "lower secondary education," "upper secondary education," "post-secondary non-tertiary education," "short-cycle tertiary

education," "bachelor's degree or equivalent," or "doctoral degree or equivalent." We recoded these to "no education," "primary," "secondary," and "higher."

- *Employment contract* (Q11): nominal variable with values equal to "indefinite," "limited," "temporary," and "apprenticeship or other training contract." We utilized only "indefinite" and "temporary" values in this study.

- *Working arrangement* (Q2d): nominal variable with values equal to "full time" and "part time."

- Occupation: nominal variable calculated according to the one-digit ISCO classification to obtain the following values: "manager," "scientific professional," "technician or associate professional," "clerical support worker," "service or sales worker," "skilled agricultural, forestry, or fishery worker," "craft or related trades worker," "plant or machine operator or assembler," "elementary occupation," and "armed forces occupation." We removed "manager" and "crafts or related trades worker" from the analysis due to a lack of observations for men and women within these values, as well as "armed forces occupation" for women.

To analyze whether the different predictor variables significantly affect telework according to gender, we applied a logistic regression model with the variable telework as the response variable and the remainder as predictors. The likelihood ratio was implemented to calculate variable significance. For the multiple comparisons, we corrected $p$-values according to the Tukey method and assumed $\alpha = 0.05$ as the level of significance. To observe how these factors affect each case, we elaborated a model for each gender and for the entire sample. Finally, we conducted evaluated the residuals of the logit model to verify the validity of the logistic regression and to determine whether the assumptions underlying the regression were fulfilled.

## 3. Results

According to the variables used to analyze telework (see Table 3), no differences were found for the life-stage variable, which included the relational variable living arrangements (cohabitation). Concerning the latter, and taking into account the variable children, the proportion of teleworkers with children and without children was very similar for men (42.4% and 39.8%, respectively) and women (56.1% and 50.0%, respectively). We found a comparable picture concerning the relational variable (living arrangements or cohabitation), as there were minimal differences among both men and women between those who telework while living alone and those who telework while living with a partner and others (40.0% and 40.9%, respectively, in the case of men; 46.7% and 53.1%, respectively, in the case of women). In contrast, there were significant differences in frequencies according to gender in the variables related to education and employment.

This can be explained as follows. First, telework tends to be more conditioned by structural factors of employment, such as sector of activity, educational level, and type of contract, than by personal or family situation. In other words, the possibility of engaging in telework depends more on the characteristics of the job than on whether an individual lives alone or with other people.

Second, the flexibility provided by telework allows both those living alone and those living with others to adapt to this modality without their living arrangements representing a significant barrier. Unlike other work dynamics that may be more influenced by household composition, telework appears to be similarly accessible to different types of households. Likewise, these results could reflect a greater diversification of profiles within telework, where personal situation does not determine individuals' participation in this type of work.

The results confirm that regarding the variable education level, telework occurs in a higher proportion among men and women with higher education (79.2% or 77.1%, respectively). This suggests that access to telework is not evenly distributed across all educational levels and is conditioned by educational background. It is possible that occupations that allow for this type of work require specialized skills or higher qualifications, or belong to sectors where the use of digital technologies is essential. In this sense, telework could be reinforcing preexisting inequalities in the labor market, favoring those with higher education and limiting access for those with lower educational levels.

**Table 3. Descriptives according to telework and life-stage, sociodemographic, and labor-related variables.**

| | Men | | | | Women | | | |
|---|---|---|---|---|---|---|---|---|
| | No | Yes | Total | Teleworking percentage | No | Yes | Total | Teleworking percentage |
| *Education level* | | | | | | | | |
| No education | 9 | 0 | 9 | 0.0 | 7 | 0 | 7 | 0.0 |
| Primary | 30 | 0 | 30 | 0.0 | 23 | 1 | 24 | 4.2 |
| Secondary | 297 | 63 | 360 | 17.5 | 205 | 72 | 277 | 26.0 |
| Higher | 54 | 205 | 259 | 79.2 | 81 | 273 | 354 | 77.1 |
| *Children* | | | | | | | | |
| No children | 254 | 168 | 422 | 39.8 | 208 | 208 | 416 | 50.0 |
| One or more children | 136 | 100 | 236 | 42.4 | 108 | 138 | 246 | 56.1 |
| *Living arrangements* | | | | | | | | |
| Alone | 66 | 44 | 110 | 40.0 | 48 | 42 | 90 | 46.7 |
| With partner and others | 324 | 224 | 548 | 40.9 | 268 | 304 | 572 | 53.1 |
| *Contract* | | | | | | | | |
| Temporary | 102 | 19 | 121 | 15.7 | 101 | 46 | 147 | 31.3 |
| Indefinite | 288 | 249 | 537 | 46.4 | 215 | 300 | 515 | 58.3 |
| *Working arrangement* | | | | | | | | |
| Part time | 44 | 10 | 54 | 18.5 | 135 | 47 | 182 | 25.8 |
| Full time | 346 | 258 | 604 | 42.7 | 181 | 299 | 480 | 62.3 |
| *Occupation* | | | | | | | | |
| Manager | 1 | 0 | 1 | 0.0 | 0 | 0 | 0 | – |
| Scientific professional | 4 | 50 | 54 | 92.6 | 2 | 41 | 43 | 95.3 |
| Technician or associate professional | 29 | 126 | 155 | 81.3 | 48 | 138 | 186 | 74.2 |
| Clerical support worker | 38 | 41 | 79 | 51.9 | 22 | 51 | 73 | 69.9 |
| Service or sales worker | 10 | 39 | 49 | 79.6 | 11 | 101 | 112 | 90.2 |
| Skilled agricultural, forestry, or fishery worker | 59 | 3 | 62 | 4.8 | 104 | 10 | 114 | 8.8 |
| Craft or related trades worker | 19 | 0 | 19 | 0.0 | 2 | 0 | 2 | 0.0 |
| Plant or machine operator or assembler | 114 | 4 | 118 | 3.4 | 20 | 3 | 23 | 13.0 |
| Elementary occupation | 69 | 2 | 71 | 2.8 | 6 | 2 | 8 | 25.0 |
| Armed forces occupation | 47 | 3 | 50 | 6.0 | 101 | 0 | 101 | 0.0 |

*Source*: EWCS, Eurofound (2021) [66].

As far as employment is concerned, particularly contract type, differences were noted in the higher proportion of employed people who telework under a contract with unlimited duration compared to those who telework under a temporary contract (46.4% and 15.7%, respectively, among men; 58.3% and 31.3%, respectively, among women). The same is true for those working full-time versus those working part-time (42.7% and 18.5%, respectively, among men; 62.3% and 25.8%, respectively, among women).

These results suggest that telework is linked not only to occupation but also to job stability and workload. People with temporary contracts and reduced working hours, who tend to be in more precarious positions, have less access to this modality, which reinforces preexisting labor inequalities. In addition, the fact that women have greater engagement in telework within these groups could be related to patterns of work–life balance and occupational segmentation, which raises questions about how telework may be reproducing gender inequalities in the labor market.

Finally, significant differences were noted in the percentages of scientific professionals engaging in telework (which were extremely high at 92.6% for men; 95.3% for women), and for mid-level technicians, administrative staff, and service or sales workers.

The results of the logistic regression analysis for both men and women (see Table 4) indicate high statistical significance for the education variable (higher education level: $p < .001$; $z = 7.959$), and the labor variable indefinite ($p < .001$; $z = 5.073$) related to contract type. The labor variable full time ($p < .05$; $z = 2.446$), linked to working arrangement, has low statistical significance. We also examined the association between the labor variable occupation type and telework, focusing on scientific professionals, which have a high probability of engaging in telework for both men and women ($0.91 \pm 0.63$ and $0.94 \pm 0.74$, respectively). All occupations under consideration have high Wald statistics and negative values, apart from service or sales worker, which does not demonstrate statistical significance.

Conversely, the relational variables children and living with a partner and others lacked statistical significance in relation to telework.

Consequently, education and labor variables were found to exert a substantial influence on telework patterns, indicating that workers with a high level of education and indefinite contracts exhibit a greater propensity to engage in telework, while those with full-time contracts demonstrate a slightly lower likelihood of doing so. On the other hand, although the labor variable related to occupation is a predictor of telework, it has an inverse character. Thus, the different occupations considered show a low propensity to engage in telework compared to the reference occupation of scientific professionals. Finally, the relational variables do not contribute to the predictive model related to telework.

The results of the logistic regression analysis distinguishing men and women (see Table 5) confirm the statistical significance of the variables related to education and the variables referred to employment (contract type, working arrangement,

**Table 4. Logistic regression model statistics summary.**

|  | Estimate Std. | Std. Error | z-Value | Pr(>|z|) |
|---|---|---|---|---|
| (Intercept) | -0.081 | 0.613 | -0.132 | .895 |
| Gender: woman | 0.102 | 0.196 | 0.519 | .604 |
| *Education level* |  |  |  |  |
| Higher | 1.593 | 0.200 | 7.959 | <.001**** |
| *Children* |  |  |  |  |
| One or more children | 0.118 | 0.198 | 0.599 | .55 |
| *Living arrangements* |  |  |  |  |
| With partner and others | 0.314 | 0.252 | 1.246 | .213 |
| *Contract* |  |  |  |  |
| Indefinite | 1.173 | 0.231 | 5.073 | <.001**** |
| *Working arrangement* |  |  |  |  |
| Full time | 0.663 | 0.259 | 2.446 | .014* |
| *Occupation* |  |  |  |  |
| Technician or associate professional | -1.716 | 0.500 | -3.431 | <.001*** |
| Clerical support worker | -2.073 | 0.513 | -4.037 | <.001**** |
| Service or sales worker | -0.410 | 0.551 | -0.743 | .457 |
| Skilled agricultural, forestry, or fishery worker | -4.449 | 0.570 | -7.806 | <.001**** |
| Plant or machine operator or assembler | -4.966 | 0.624 | -7.953 | <.001**** |
| Elementary occupation | -4.979 | 0.714 | -6.978 | <.001**** |
| Armed forces occupation | -5.738 | 0.766 | -7.489 | <.001**** |

*Note*: 0 ****.0001 ***.001 **.01 *.05..1 ns 1. AIC = 832.98.

*Source*: Own elaboration based on EWCS, Eurofound (2021) [66].

and occupation type), while having one or more children and living with a partner and others do not show significance. In other words, the relational variable living arrangements (cohabitation) lacks statistical significance linked to telework.

In the case of men, higher education level exhibits statistical significance ($p < .001$) and a high Wald statistic (6.031), meaning that it contributes significantly to the propensity to engage in telework within the predictive model. The same is true for the indefinite duration contract type ($p < .001$; $z = 4.723$). In this sense, several of the occupational variables stand out, since scientific professionals, which were used as a reference for this analysis, were found to have a high propensity to engage in telework (0.91 ± 0.63). This is also the case for clerical workers ($p < .001$; $z = -3.106$), skilled agricultural, forestry, and fishery workers ($p < .001$; $z = -5.559$), plant machine operators and assemblers ($p < .001$; $z = -6.448$), those in elementary occupations ($p < .001$; $z = -5.725$), and those in the armed forces ($p < .001$; $z = -5.018$), all of which have high but negative Wald statistics and contribute significantly to the predictive modeling related to telework. Nevertheless, this contribution is inverse as such occupations have a low propensity to engage in telework, particularly when compared to scientific professionals.

In the case of women, a significant relation was also noted among the variables higher education level ($p < .001$; $z = 5.200$) and contract with indefinite duration ($p < .001$; $z = -2.761$), and the occupational variables clerical worker ($p < .05$; $z = -2.384$), skilled agricultural, forestry, or fishery worker ($p < .001$; $z = -5.117$), plant or machine operator or assembler ($p < .001$; $z = -4.011$), and elementary occupation ($p < .001$; $z = -3.060$).

Several differences were noted between gender regarding the labor variables. First, the indefinite contract type exhibits less statistical significance for women ($p < .01$; $z = 2.761$) versus men ($p < .001$; $z = 4.723$). This indicates that having an indefinite contract and engaging in telework is more likely for men than for women. Second, the variable full time, related to working arrangement, shows statistical significance for women ($p < .05$; $z = 2.092$) but not for men. This indicates that

**Table 5. Logistic regression model statistics summary by gender.**

| | Men | | | | Women | | | |
|---|---|---|---|---|---|---|---|---|
| | Estimate Std. | Std. Error | z-Value | Pr(>\|z\|) | Estimate Std. | Std. Error | z-Value | Pr(>\|z\|) |
| (Intercept) | -1.044 | 0.948 | -1.102 | .2704 | 0.331 | 0.883 | 0.375 | .7074 |
| *Education level* | | | | | | | | |
| Higher | 1.755 | 0.291 | 6.031 | <.001**** | 1.548 | 0.298 | 5.200 | <.001**** |
| *Children* | | | | | | | | |
| One or more children | 0.047 | 0.313 | 0.149 | .8813 | 0.175 | 0.270 | 0.649 | .5166 |
| *Living arrangements* | | | | | | | | |
| With partner and others | 0.398 | 0.366 | 1.086 | .2775 | 0.313 | 0.363 | 0.863 | .3882 |
| *Contract* | | | | | | | | |
| Indefinite | 1.828 | 0.387 | 4.723 | <.001**** | 0.819 | 0.297 | 2.761 | .0058** |
| *Working arrangement* | | | | | | | | |
| Full time | 0.786 | 0.537 | 1.465 | .1428 | 0.643 | 0.307 | 2.092 | .0365* |
| *Occupation* | | | | | | | | |
| Technician or associate professional | -1.206 | 0.676 | -1.784 | .0744 | -2.003 | 0.764 | -2.622 | .0087** |
| Clerical support worker | -2.126 | 0.684 | -3.106 | .0019** | -1.888 | 0.792 | -2.384 | .0171* |
| Service or sales worker | -0.986 | 0.754 | -1.308 | .191 | -0.098 | 0.826 | -0.119 | .9056 |
| Skilled agricultural, forestry, or fishery worker | -4.906 | 0.882 | -5.559 | <.001**** | -4.232 | 0.827 | -5.117 | <.001**** |
| Plant or machine operator or assembler | -5.268 | 0.817 | -6.448 | <.001**** | -4.059 | 1.012 | -4.011 | <.001**** |
| Elementary occupation | -5.539 | 0.968 | -5.725 | <.001**** | -3.449 | 1.127 | -3.060 | .0022** |
| Armed forces occupation | -4.469 | 0.891 | -5.018 | <.001**** | – | – | – | – |

*Note*: 0 ****.0001 ***.001 **.01 *.05.1 1. AIC = 450.58.

*Source*: Own elaboration based on EWCS, Eurofound (2021) [66].

the combination of full-time employment and engaging in telework is more prevalent among women than among men. Finally, both clerical support worker and elementary occupation exhibit lower degrees of statistical significance in the case of women compared to men. Thus, in these occupations, women exhibit a lower propensity to engage in telework compared to men. Nevertheless, it is noteworthy that the variable technician or associate professional, related to occupation, is statistically significant only for women ($p < .01$; $z = -2.622$), suggesting that those engaged in such occupations are more likely to engage in telework compared to their male counterparts.

Table 6 presents the ratios and confidence intervals of the variables considered in the telework predictor model, considering both men and women. The findings indicate that the variables higher education level, indefinite contract type, and full-time working arrangement have odds ratios (ORs) greater than 1. This suggests that these aspects directly influence the propensity to engage in telework. In addition, the strength of the relationship is more robust in the case of the higher education level variable (OR = 5.03) compared to the indefinite contract type variable (OR = 3.35) and the full-time working arrangement variable (OR = 1.76).

Conversely, the labor variables related to occupation exhibit ORs of less than 1, meaning that their influence on telework is inverse. Especially significant is the strength of the inverse relationship between telework and the occupations skilled agricultural, forestry, or fishery worker; plant or machine operator or assembler; elementary occupation; and armed forces occupation. This indicates that the propensity to engage in telework in these occupations is very low.

Finally, the test of multicollinearity, based on the generalized variance inflation factor (GVIF), indicates that there are no collinearity problems in this study, as all GVIF values are close to 1.

Table 7 shows the ratios and CIs of the variables in the predictive modeling that allowed us to forecast the propensity to engage in telework, considering gender-specific samples. The table shows that the ORs of the variables higher education and indefinite contract type are higher than 1 in the case of men and, consequently, that their influence on telework

**Table 6. Confidence intervals and multicollinearity according to GVIF.**

| | OR | 95% | CI[1] | *p*-Value | GVIF | GVIF[2] |
|---|---|---|---|---|---|---|
| *Education level* | | | | <.001 | 1.2 | 1.1 |
| Secondary | | | | | | |
| Higher | 5.03 | 3.42 | 7.42 | | | |
| *Contract* | | | | <.001 | 1.0 | 1.0 |
| Temporary | | | | | | |
| Indefinite | 3.35 | 2.15 | 5.26 | | | |
| *Working arrangement* | | | | .026 | 1.1 | 1.0 |
| Part time | | | | | | |
| Full time | 1.76 | 1.07 | 2.88 | | | |
| *Occupation* | | | | <.001 | 1.2 | 1.0 |
| Scientific professional | | | | | | |
| Technician or associate professional | 0.18 | 0.06 | 0.44 | | | |
| Clerical support worker | 0.13 | 0.04 | 0.33 | | | |
| Service or sales worker | 0.71 | 0.22 | 1.94 | | | |
| Skilled agricultural, forestry, or fishery worker | 0.01 | 0.00 | 0.03 | | | |
| Plant or machine operator or assembler | 0.01 | 0.00 | 0.02 | | | |
| Elementary occupation | 0.01 | 0.00 | 0.02 | | | |
| Armed forces occupation | 0.00 | 0.00 | 0.01 | | | |

[1]CI = confidence interval.

[2]GVIF^[1/(2*df)].

*Source*: Own elaboration based on EWCS, Eurofound (2021) [66].

is direct. In contrast, labor-related variables referring to occupation have ORs below 1, and thus have an inverse relation with telework.

The results for the sample of women are similar, since the variables higher education and permanent contract have ORs greater than 1, while the labor variables referring to occupation have ORs lower than 1. First, full-time work has an OR higher than 1 in the case of women, and is therefore a descriptor to be added in the explanation of female telework, while it is not relevant in the case of men. Second, the variable indefinite contract type exhibits a stronger association with telework among men (OR = 6.70) compared to women (OR = 2.44). Third, higher education level emerges as a more robust predictor for men (OR = 5.75) compared to women (OR = 4.85). Finally, in the context of the variables associated with occupation, it is noteworthy that the OR for the occupation service or sales worker is 0.35 for men, while for women it is 0.90, indicating that its effect on female telework is virtually negligible.

Finally, the test of multicollinearity, based on the GVIFs, indicates that there are no collinearity problems as the values are all close to 1.

Figs 2 and 3 show the propensity to engage in telework for men compared to women, considering the variables related to occupation. In general terms, we can divide occupations into two groups: the upper group comprises occupations with the highest propensity to engage in telework, which includes individuals with high qualifications linked to knowledge and those linked to services (scientific professionals, technicians and associate professionals, clerical support workers, and service or sales workers), bearing in mind that scientific professionals have the highest propensity to engage in telework among both men (0.91 ± 0.63) and women (0.94 ± 0.74). The other group includes individuals in occupations with a very low propensity to engage in telework and that are low-skilled (skilled agricultural, forestry, or fishery workers; plant or machine operators or assemblers; those in elementary occupations; and those in the armed forces).

**Table 7. CIs and multicollinearity according to variance inflation factors, by gender.**

| | Men | | | | | | Women | | | | | |
|---|---|---|---|---|---|---|---|---|---|---|---|---|
| | OR[1] | 95% | CI | *p*-Value | GVIF | GVIF[1] | OR | 95% | CI | *p*-Value | GVIF | GVIF[1] |
| *Education level* | | | | <.001 | 1.1 | 1.0 | | | | <.001 | 1.3 | 1.2 |
| Secondary | – | – | – | | | | – | – | – | | | |
| Higher | 5.75 | 3.28 | 10.2 | | | | 4.85 | 2.73 | 8.74 | | | |
| *Contract* | | | | <.001 | 1.1 | 1.0 | | | | .002 | 1.0 | 1.0 |
| Temporary | – | – | – | | | | – | – | – | | | |
| Indefinite | 6.70 | 3.22 | 14.4 | | | | 2.44 | 1.38 | 4.34 | | | |
| *Working arrangement* | | | | | | | | | | .072 | 1.1 | 1.0 |
| Part time | | | | | | | – | – | – | | | |
| Full time | | | | | | | 1.72 | 0.95 | 3.07 | | | |
| *Occupation* | | | | <.001 | 1.1 | 1.0 | | | | <.001 | 1.4 | 1.0 |
| Scientific professional | – | – | – | | | | – | – | – | | | |
| Technician or associate professional | 0.27 | 0.06 | 0.87 | | | | 0.13 | 0.02 | 0.48 | | | |
| Clerical support worker | 0.11 | 0.02 | 0.38 | | | | 0.15 | 0.02 | 0.60 | | | |
| Service or sales worker | 0.35 | 0.07 | 1.39 | | | | 0.90 | 0.13 | 3.85 | | | |
| Skilled agricultural, forestry, or fishery worker | 0.01 | 0.00 | 0.03 | | | | 0.01 | 0.00 | 0.06 | | | |
| Plant or machine operator or assembler | 0.00 | 0.00 | 0.02 | | | | 0.02 | 0.00 | 0.09 | | | |
| Elementary occupation | 0.00 | 0.00 | 0.02 | | | | 0.03 | 0.00 | 0.22 | | | |
| Armed forces occupation | 0.01 | 0.00 | 0.05 | | | | – | – | – | | | |

[1]GVIF^[1/(2*df)].

*Source*: Own elaboration based on EWCS, Eurofound (2021) [66].

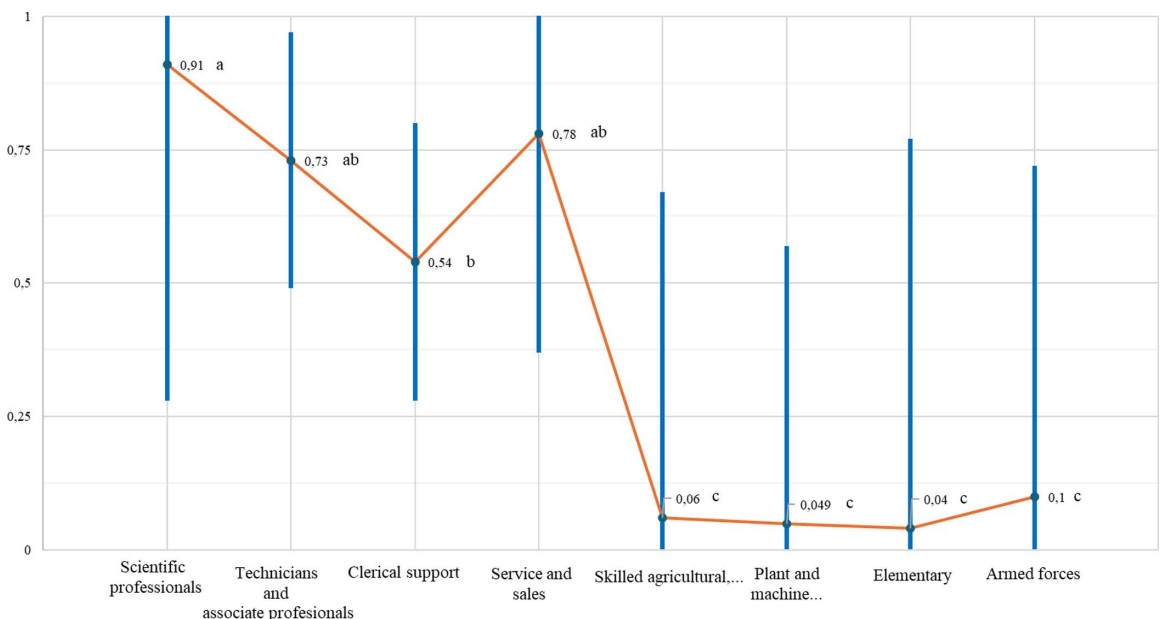

**Fig 2. Propensity to engage in telework among men according to occupation.**

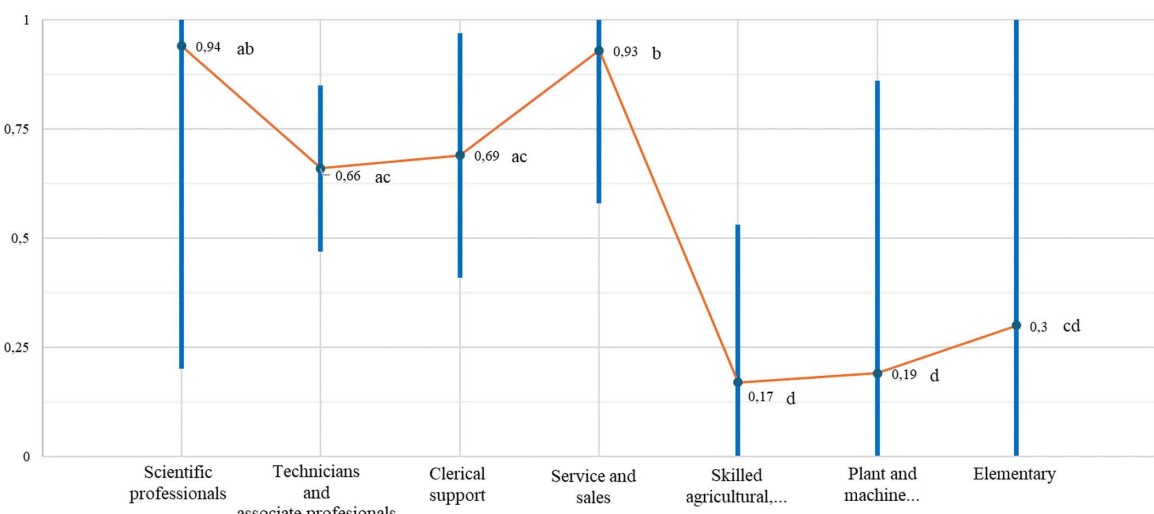

**Fig 3. Women's propensity to engage in telework according to occupation.** Source: Own elaboration based on EWCS, Eurofound (2 021) [66].

Regarding the propensity to engage in telework according to gender more specifically, Fig 2 shows the significance of differences in the probability of men engaging in telework according to occupation by using the CLD method. This method assigns letters to each level so that only the levels that have different letters have significant differences. We identified the existence of two groups: the upper group (comprising subgroups a, ab, and b) includes scientific professionals (a; 0.91±0.63) with a very high propensity to engage in telework; technicians and associate professionals (ab; 0.73±0.24), and service or sales workers (ab; 0.78±0.41), with a high propensity to engage in telework; and clerical support workers (b; 0.54±0.26), with a medium propensity to engage in telework. The lower group (c) includes occupations with very low

or almost zero propensity to engage in telework, such as skilled agricultural, forestry, or fishery workers (c; 0.06±0.61); plant or machine operators or assemblers (c; 0.049±0.52); those in elementary occupations (c; 0.04±0.73); and those in the armed forces (c; 0.1±0.62).

Fig 3 shows significant differences in the propensity to engage in telework for women depending on their occupation. Again, we identified two groups: the upper group (comprising subgroups ab, ac, and b) includes scientific professionals (ab; 0.94±0.74) and service or sales workers (b; 0.93±0.35), with a very high propensity to engage in telework; and technicians and associate professionals (ac; 0.66±0.19), and clerical support workers (ac; 0.69±0.28), with a high propensity to engage in telework. The lower group (comprising subgroups d and cd) includes skilled agricultural, forestry, or fishery workers (d; 0.17±0.36); plant or machine operators or assemblers (d; 0.19±0.67); and those in elementary occupations (cd; 0.3±0.86), with a low propensity to engage in telework. Due to the large error margin, this group represents an anomaly as it does not differ significantly from the occupations in the lower group nor those with higher-level occupations, such as technicians and associate professionals, and clerical support workers (ac).

Thus, the results reveal manifest gender differences. In particular, women, in contrast to men, who focus on higher-skilled and knowledge-related activities, are prone to engage in telework in all occupations, including lower-skilled and medium-skilled ones, such as service or sales workers.

Finally, we evaluated the residuals of the logit model (Fig 4). This showed that the residuals behave in accordance with the expected values, with no deviations, absence of collinearity, or poor model fit.

The results support H2, which proposed that occupation is a moderating factor in gender-based differences determining the propensity to engage in telework. This finding aligns with those of Asmussen *et al.* (2024) [31]; Chung and van der Lippe (2020) [53]; Gálvez *et al.* (2021) [75]; Kley and Reimer (2023) [41]; Minkus *et al.* (2022) [73]; López-Igual and Rodríguez-Modroño (2020) [76]; Lyttelton *et al.* (2020) [68], and Sostero *et al.* (2020) [62]. In this case, we confirm the establishment of gender discrimination that links telework to women's occupations at all skill levels.

H3 is also supported—that is, education moderates the propensity to engage in telework for both men and women. This again aligns with other studies [31,41,72,76,77]. In this sense, higher education, as a moderating factor that affects the propensity to engage in telework, is especially noteworthy because of its positive influence.

The results do not support H1, which suggested that household composition is a moderator that affects the relationship between gender and the propensity to engage in telework. Despite prior findings on this issue [15,76,79], we found no significant relationship for the variable children or the relational variable living arrangements (cohabitation) since, in both cases, the propensity to engage in telework among men versus women was found to be very similar.

## 4. Discussion

The results of the analysis show that the occupational and educational variables have a greater impact on the propensity to engage in telework than the life-stage variable, which includes the relational variable living arrangements (cohabitation). This finding provides two main insights.

First, as a consequence of technology shift and labor market, higher-skilled jobs (professional and technical jobs) are the most susceptible to operate with the formulas that telework demands [15,68,80–82]. In contrast, there is less possibility to engage in telework for simple, manual activities or services, except in administrative and some commercial areas [80,81,83]. Therefore, the nature of the occupation is a determinant of engaging in telework [81–83]. Our findings confirm that in Spain there is also a higher propensity to engage in telework in high-skilled occupations linked to proficiency, and in the service sector. Moreover, in the case of the Spanish labor market, we found that women's propensity to engage in telework is high in all occupations, not only in those that are highly skilled and linked to knowledge but also in those requiring lesser qualifications, such as services or sales, which is a feminized labor niche.

The second insight emphasizes the flexibilization of employment forms, especially those linked to telework. Such flexibilization is determined by the labor market's occupational transformation, regardless of employee needs or preferences

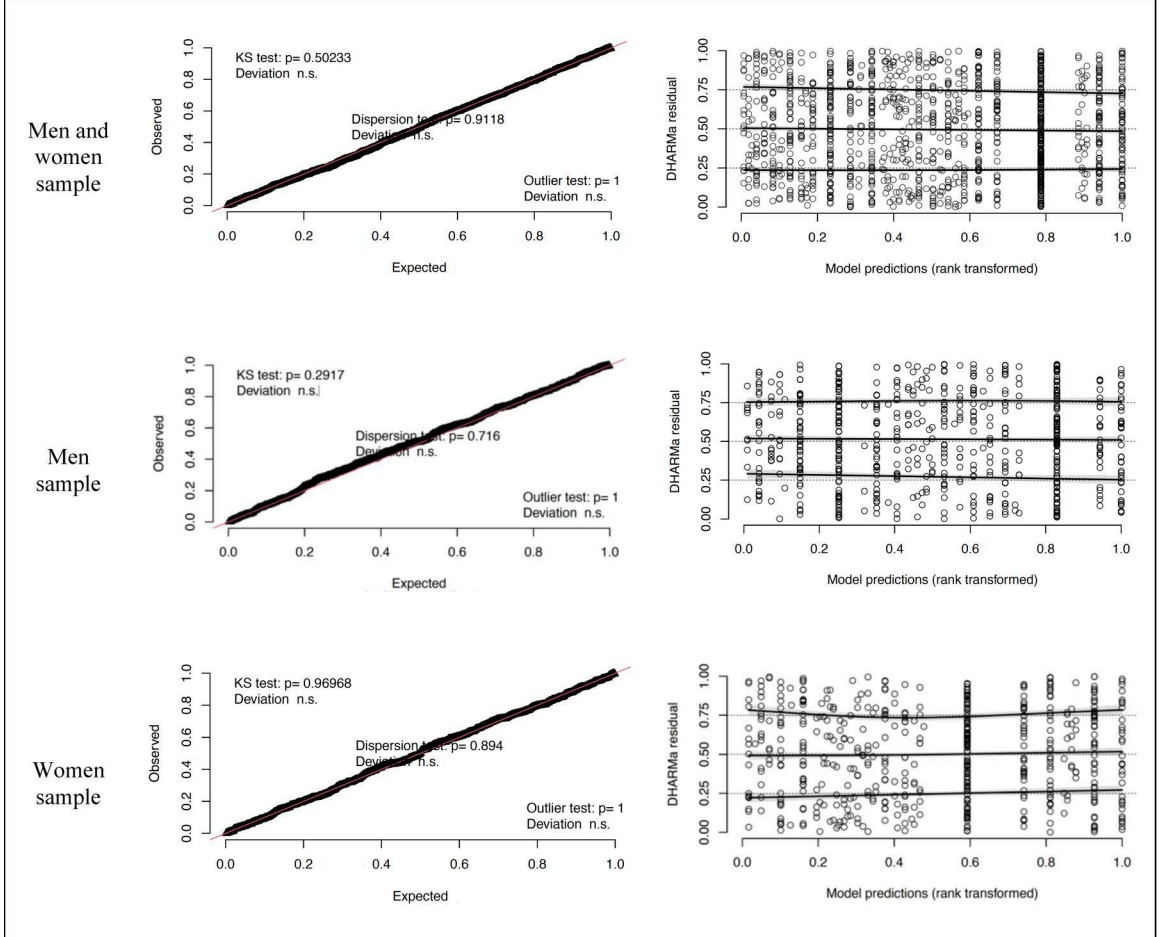

**Fig 4. Logit model residual analysis.** Source: Own elaboration based on EWCS, Eurofound (2021) [66].

related to work–life balance. In this sense, research has often indicated that telework does not change gender roles but rather reinforces traditional roles, since it allows women to both conduct household tasks and work [19,84–86]. This devalues telework and, in some cases, its degradation into a flexible form of employment [87]. Following other studies [41,76,88], we found that the flexibility that results from telework is compatible with women's full-time work, as it has been found. However, independently of gender roles, telework is becoming increasingly prevalent in specific occupational categories among both men and women, which indicates that, to a significant extent, propensity to engage in telework is determined by decisions based on work-related criteria, rather than those concerning family or work–life balance.

Regarding the post-pandemic period, occupation, contract type, and working arrangement, as employment-related factors whose values did not indicate any relevant gender differences in this study, were found to be the best predictors of the propensity to engage in telework among both genders. This shows that issues such as flexibility or digitalization, especially in skilled occupations, are more impactful than variables linked to the traditional gender roles characterizing a classic approach.

This finding is consistent with the fact that the number of occupations that professionals can perform remotely has increased across the EU since the outbreak of the pandemic. The sector with the most roles that can be conducted remotely is services (93%), followed by information and communications (79%), education (68%), and other professional, scientific, and technical activities (66%). Regarding occupation, the proportion of professionals who could technically work

from home comprises more than 70% of managers and professionals, 50% of technicians, and 83% of administrative workers who previously had very limited access to telework [89].

In contrast, telework possibilities in medium and low-skilled occupations are more limited, as the type of activity or technology does not make such occupations compatible with working remotely. Occupational segregation also explains why a higher percentage of women engage in telework. While men tend to be overrepresented in sectors with limited potential for telework (e.g., manufacturing and construction), women more often hold occupations where telework is possible, such as administrative tasks, even if they work within, for example, the manufacturing and construction sector [89].

The results suggest that the gender divide with respect to labor in general explains why there are differences in the participation of men versus women in telework. In this sense, we found that the reason why such participation is higher among women than men lies in the overrepresentation of women in occupations where it is possible to telework, rather than in a drive to obtain work–life balance.

We found that the occupational dimension is linked to the educational one, as a high education level is a predictor of telework, per previous research [15,76,88,90]. This is also consistent with the finding that there is a high likelihood of engaging in telework in medium- to high-skill professions, such as scientific occupations, which also entail an increased tendency to engage in telework [76,81,88].

## 5. Conclusions

This research found that work–life balance is not a significant predictor of propensity to engage in telework when considering dimensions related to cohabitation and children. However, propensity to engage in telework is strongly influenced by gender. In this sense, a report on gender indicators by the Digital Society [91] indicated that in the digital context, the COVID-19 pandemic strengthened preexisting gender inequalities in Spain, and the EU in general, as it meant that women began performing more activities related to family care, communication, health, and education, while engaging in telework. Based on the report, telework can deteriorate the working and psychosocial conditions of women by reinforcing the inequality linked to traditional gender roles. These findings are consistent with research by Haddon and Brynin (2005) [51] and Wheatley (2012) [50].

The results of this research can serve as a basis for the formulation of more equitable labor policies adapted to the needs of workers at different stages of their professional and personal lives. Although teleworking is regulated in detail in Spain as *Ley 10/2021, de 9 de julio, de trabajo a distancia* [64], studies such as this one can provide evidence for the revision of regulations on labor flexibility, work-life balance, and the protection of workers' rights in teleworking contexts. It is necessary to bear in mind that regulations must adapt to the changing reality of employment formulas, so research such as this can help to inform regulatory adaptations on the subject. They can also guide the design of measures to reduce gender inequalities in the access and use of telework, ensuring that this does not deepen pre-existing gaps in the labor market.

The question is whether this form of employment contributes to the persistence of gender inequalities. Given the evolution of telework in Europe, it could be argued that this is the case in Spain and other countries, the growth of this work modality since the pandemic has reinforced gender roles, which are still strongly marked by traditionalism. Thus, despite the positive outcomes of telework as shown in various studies, there is a need to ensure that it does not become an additional element of discrimination against women in the labor market.

Therefore, while it appears that telework is experiencing growth, it is essential to look to the future and continue to identify both its strengths and threats, especially if is reinforcing preexisting stereotypes or practices. In this regard, there is a need to promote actions that create new opportunities for all individuals, with a particular focus on achieving gender equality in the workplace. This involves not only removing obstacles that hinder women's participation in economic activities but also implementing social protection measures and closing gaps in professional environments, ensuring fair and equitable access to telework opportunities.

This research opens interesting avenues of work that would deepen the understanding of telework as a social and labor phenomenon. In particular, it would be useful to explore in more detail the unequal development of telework from a gender perspective, considering the differences that may arise depending on the type of work arrangement (full vs. part time). Future research could also delve deeper into how work dynamics in telework environments differently affect men and women, examining possible structural inequalities and specific patterns that have not been addressed in depth in this paper. Likewise, it would be valuable to investigate telework experiences according to gender in different occupations using a qualitative methodology.

Another valuable line of research would be to assess how these gender differences in telework relate to family composition, particularly in households with children. In this sense, it could be of particular interest to study the impact of telework as a function of the age of the employee's children, since care needs and family organization vary significantly according to this variable. This approach would make it possible to identify more precisely how telework influences family and work responsibilities, and how these interactions may be a key factor in perpetuating or reducing gender inequalities in the work and domestic spheres.

## Author contributions

**Conceptualization:** Pilar Ortiz Garcia, Salvador Manzanera-Román.

**Data curation:** Pilar Ortiz Garcia.

**Investigation:** Pilar Ortiz Garcia.

**Methodology:** Pilar Ortiz Garcia, Salvador Manzanera-Román.

**Resources:** Salvador Manzanera-Román.

**Writing – original draft:** Pilar Ortiz Garcia, Salvador Manzanera-Román.

**Writing – review & editing:** Pilar Ortiz Garcia, Salvador Manzanera-Román.

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
