## [Decision Letter · Decision Letter 0]

16 Jan 2025

PONE-D-24-47164Importance of occupation in the increase of gender differences in the advance of telework in SpainPLOS ONE

Dear Dr. Salvador Manzanera-Román,

Thank you for submitting your manuscript to PLOS ONE. After careful consideration, we feel that it has merit but does not fully meet PLOS ONE’s publication criteria as it currently stands. Therefore, we invite you to submit a revised version of the manuscript that addresses the points raised during the review process.

The suggested changes that must be added to the revised version of the paper before it can be accepted for publication in PLOS ONE are detailed next.

The manuscript's analytical approach is too descriptive. It must explain in more detail the causal relationships that affect telework adoption, such as between teleworking and the proposed affecting conditions.

There is a visible lack of figures and statistics related to the case of Spain. Therefore, a table showing the state of telework in Spain during the study's analysis period must be added.

The proposed hypotheses must be formulated more explicitly to control for the proposed influential variables.

Although the logit model appears suitable for the paper's purposes, presenting marginal effects could offer a more intuitive result interpretation. Besides, additional diagnostic tests for logistic regression assumptions, discussion of measurement scales, and the use of sample weights must be added.

The reduction in the sample size from 2,903 to 1,320 individuals must be adequately explained.

The empirical work needs to present results for the entire and gender-specific samples.

Some results in Tables 3 and 4 show striking gender differences that require further explanation.

The study's timing during the pandemic should be emphasized by specifying when the effects were recorded.

The construction of the dichotomous variable for teleworking must be precisely explained.

The general presentation must be improved. Revise manuscript sections, complete tables with standard logistic regression statistics, and include titles and sources for graphs.

The paper's limitations must be mentioned by adding comments to acknowledge the study's constraints.

The conclusion section must be refined by separating and strengthening it, ensuring it reflects the refined analysis and addresses policy implications based on the findings.

Finally, the manuscript must homogenize the use of American English in the text, such as substituting the word "labor" instead of "labour" and using decimal points instead of commas to describe numerical data.

We look forward to receiving your revised manuscript.

Kind regards,

Humberto Merritt, PhD

Academic Editor

PLOS ONE

Journal Requirements:

2. Please update your submission to use the PLOS LaTeX template. The template and more information on our requirements for LaTeX submissions can be found at http://journals.plos.org/plosone/s/latex .

3. Please include your tables as part of your main manuscript and remove the individual files. Please note that supplementary tables (should remain/ be uploaded) as separate "supporting information" files.

4. Please note that your Data Availability Statement is currently missing [the repository name and/or the DOI/accession number of each dataset OR a direct link to access each database]. If your manuscript is accepted for publication, you will be asked to provide these details on a very short timeline. We therefore suggest that you provide this information now, though we will not hold up the peer review process if you are unable.

Comments from PLOS Editorial Office: We note that one or more reviewers has recommended that you cite specific previously published works. As always, we recommend that you please review and evaluate the requested works to determine whether they are relevant and should be cited. It is not a requirement to cite these works. We appreciate your attention to this request.

Additional Editor Comments (if provided):

The manuscript provides valuable insights into teleworking patterns in Spain, particularly regarding gender differences and household composition. The research demonstrates that occupation, contract type, and working time are the best predictors of telework probability for both men and women. This indicates that job flexibility and digitalization factors, especially in skilled occupations, are more explanatory than variables traditionally associated with gender roles. Identifying occupation-specific telework patterns for women is a relevant contribution to the field. Yet, the study could have provided a comparative analysis with other EU countries to get additional insights into the uniqueness of the Spanish case. Besides, investigating the quality of telework experiences across different occupations and genders would have also provided valuable insights.

On the other hand, several shortcomings must be corrected before the paper can be accepted for publication. In this respect, apart from the observations posed by the paper's reviewers, additional drawbacks need to be addressed.

For example, the paper could have benefited from more rigorous methodological approaches, more precise variable definitions, and a deeper exploration of the causal relationships regarding telework adoption. Although the manuscript proposes three axes of analysis to explain the telework phenomenon in Spain (i.e., productive strategy, labor relations, and psychosocial consequences), there is a very superficial academic analysis of teleworking and its evolution in the wake of the COVID-19 pandemic. Moreover, despite the extensive list of references, there is a notable lack of details, statistics, and comments regarding telework in Spain. For example, the literature review is notably weak and dated because it rests on too many old references, where a quarter of the 68 entries are more than ten years old, with 17 including the COVID-19 issue in their analyses and only one referring to the particular case of Spain. Therefore, additional references must be added to further the purpose of the study. To help address these limitations, a short list of pertinent references is listed next.

Chowhan, J., MacDonald, K., Mann, S. L., & Cooke, G. B. (2021). Telework in Canada: Who Is Working from Home during the COVID-19 Pandemic? Industrial Relations, 76(4): 761-791. https://doi.org/10.7202/1086009ar

Collins, C., Landivar, L. C., Ruppanner, L., & Scarborough, W. J. (2021). COVID-19 and the Gender Gap in Work Hours. Gender, Work & Organization, 28(S1): 101-112. https://doi.org/10.1111/gwao.12506

Grubanov, S., Spyratos, S., & Iacus, S. M. (2022). Monitoring COVID-19-Induced Gender Differences in Teleworking Rates Using Mobile Network Data. Journal of Data Science, 20(2): 209-227. https://doi.org/10.6339/22-JDS1043

Hayes, S. W., Priestley, J. L., Moore, B. A., & Ray, H. E. (2021). Perceived Stress, Work-Related Burnout, and Working From Home Before and During COVID-19: An Examination of Workers in the United States. Sage Open, 11(4): 1-12. https://doi.org/10.1177/21582440211058193

Karjalainen, M. (2023). Gender and the Blurring Boundaries of Work in the Era of Telework—A Longitudinal Study. Sociology Compass, 17(1): e13029. https://doi.org/10.1111/soc4.13029

Minkus, L., Groepler, N., & Drobnič, S. (2022). The Significance of Occupations, Family Responsibilities, and Gender for Working from Home: Lessons from COVID-19. PLoS ONE, 17(6): e0266393. https://doi.org/10.1371/journal.pone.0266393

Okubo, T. (2022). Telework in the Spread of COVID-19. Information Economics and Policy, 60(100987): 1-16. https://doi.org/10.1016/j.infoecopol.2022.100987

Pabilonia, S. W., & Vernon, V. (2022). Telework, Wages, and Time Use in the United States. Review of Economics of the Household, 20(3): 687-734. https://doi.org/10.1007/s11150-022-09601-1

Raisiene, A. G., Rapuano, V., Varkuleviciute, K., & Stachová, K. (2020). Working from Home—Who Is Happy? A Survey of Lithuania's Employees during the COVID-19 Quarantine Period. Sustainability, 12(13): 1-21. https://doi.org/10.3390/su12135332

Schieman, S., Badawy, P. J., Milkie, M. A., & Bierman, A. (2021). Work-Life Conflict During the COVID-19 Pandemic. Socius, 7: 1-19. https://doi.org/10.1177/2378023120982856

Tahlyan, D., Said, M., Mahmassani, H., Stathopoulos, A., et al. (2022). For Whom Did Telework Not Work During the Pandemic? Understanding the Factors Impacting Telework Satisfaction in the US Using a Multiple Indicator Multiple Cause (MIMIC) Model. Transportation Research Part A: Policy and Practice, 155: 387-402. https://doi.org/10.1016/j.tra.2021.11.025

Touzet, C. (2023). Teleworking through the Gender Looking Glass: Facts and Gaps. Paris: OECD. February 2023. https://dx.doi.org/10.1787/8aff1a74-en

Vandecasteele, L., Ivanova, K., Sieben, I., & Reeskens, T. (2022). Changing Attitudes about the Impact of Women's Employment on Families: The COVID-19 Pandemic Effect. Gender, Work & Organization, 29(6): 2012-2033. https://doi.org/10.1111/gwao.12874

Wels, J., & Hamarat, N. (2021). A Shift in Women's Health? Older Workers' Self-Reported Health and Employment Settings During the COVID-19 Pandemic. European Journal of Public Health, 32(1): 80-86. https://doi.org/10.1093/eurpub/ckab204

Zoch, G., Bächmann, A. C., & Vicari, B. (2022). Reduced Well‐Being During the COVID‐19 Pandemic – The Role of Working Conditions. Gender, Work & Organization, 29(6): 1969-1990. https://doi.org/10.1111/gwao.12777

Another point concerns the comment that COVID-19 has altered the digitalization process of modern societies; however, there is a visible lack of discussion, comments and criticisms on how the pandemic has changed that process.

The suggested changes that must be added to the revised version of the paper before it can be accepted for publication in PLOS ONE are detailed next.

The manuscript's analytical approach is too descriptive. It must explain in more detail the causal relationships that affect telework adoption, such as between teleworking and the proposed affecting conditions.

There is a visible lack of figures and statistics related to the case of Spain. Therefore, a table showing the state of telework in Spain during the study's analysis period must be added.

The proposed hypotheses must be formulated more explicitly to control for the proposed influential variables.

Although the logit model appears suitable for the paper's purposes, presenting marginal effects could offer a more intuitive result interpretation. Besides, additional diagnostic tests for logistic regression assumptions, discussion of measurement scales, and the use of sample weights must be added.

The reduction in the sample size from 2,903 to 1,320 individuals must be adequately explained.

The empirical work needs to present results for the entire and gender-specific samples.

Some results in Tables 3 and 4 show striking gender differences that require further explanation.

The study's timing during the pandemic should be emphasized by specifying when the effects were recorded.

The construction of the dichotomous variable for teleworking must be precisely explained.

The general presentation must be improved. Revise manuscript sections, complete tables with standard logistic regression statistics, and include titles and sources for graphs.

The paper's limitations must be mentioned by adding comments to acknowledge the study's constraints.

The conclusion section must be refined by separating and strengthening it, ensuring it reflects the refined analysis and addresses policy implications based on the findings.

Finally, the manuscript must homogenize the use of American English in the text, such as substituting the word "labor" instead of "labour" and using decimal points instead of commas to describe numerical data.

Due to these conditions, the editor's decision for the submitted manuscript is that it needs a major revision of the whole paper.

To this effect, you have 45 days to address these comments, requiring resubmission of the corrected version to Journal PLOS ONE.

Reviewers' comments:

Reviewer's Responses to Questions

**Comments to the Author**

1. Is the manuscript technically sound, and do the data support the conclusions?

Reviewer #1: No

Reviewer #2: Yes

2. Has the statistical analysis been performed appropriately and rigorously? 

Reviewer #1: No

Reviewer #2: No

3. Have the authors made all data underlying the findings in their manuscript fully available?

Reviewer #1: Yes

Reviewer #2: Yes

4. Is the manuscript presented in an intelligible fashion and written in standard English?

Reviewer #1: Yes

Reviewer #2: Yes

5. Review Comments to the Author

Reviewer #1: 1. Research Hypotheses

The article analyzes the factors influencing the likelihood of teleworking in Spain. The three research hypotheses proposed are somewhat simplistic. While they allow for a descriptive examination of how household composition, occupation, and education affect the probability of teleworking, it would be advisable to go further and formulate more sophisticated hypotheses. These could consider the necessity of controlling for household composition, educational attainment, and occupation—variables that common sense suggests are influential in teleworking.

2. Methodology

The use of the logit model is suitable for describing the variables associated with teleworking. As a suggestion, presenting marginal effects could offer a more intuitive interpretation of the results.

3. Descriptive Tone

The overall tone of the article is somewhat descriptive. Could the analysis have explored causal relationships, for example, between teleworking and job satisfaction (or stress, isolation, etc.)? Does the survey include data on these or other variables (e.g., burnout, well-being, work-life balance)? The literature review addresses many of these issues, which are not subsequently investigated in the empirical work. For instance, the review mentions potential increases in job satisfaction due to better time management or reductions in opportunities for social relationships. Later, it is stated that teleworking may deteriorate working conditions. Investigating satisfaction levels could offer insights into whether teleworking is associated with deteriorated job satisfaction, working conditions, or work-life balance.

4. Selection Bias and Causal Effects

If the authors consider studying the causal effect of teleworking on satisfaction or work-life balance, it is recommended to control for selection bias. Men and women tend to work in different occupations, with men overrepresented in manual jobs and women in white-collar jobs.

5. Sample Size Reduction

The drastic reduction in sample size from 2,903 observations in the original database to only 1,320 individuals in the final sample is not well-explained. The article refers to the inclusion of relevant variables for analysis and an (unjustified) selection of specific Spanish data.

6. Gender-Specific Estimates

The empirical work could have presented estimation results for the entire sample (men and women) in addition to the two gender-specific samples, including a dummy variable to capture the effect of being female on the likelihood of teleworking.

7. COVID-19 Context

It should be emphasized that 2021 was still heavily influenced by the COVID-19 pandemic (the second major wave occurred in 2022).

8. Definition of Key Variables

The definition of the variable of interest is not precisely explained. While the article mentions the source variable, it does not detail how the dichotomous variable was constructed. It refers to working with a variable created by Eurofound (Full-time teleworking, Part-time teleworking, Occasional teleworking, and On-site work with some degree of teleworkability). Were all categories used or only some of them? It might be simpler to construct the dummy variable from the original variable that asks: "How often have you worked in the following places? YOUR OWN HOME (Never, Rarely, Sometimes, Often, Always)."

9. Descriptive Tables

Some results from the descriptive tables are striking. For example, in Table 2, there are significant gender differences in teleworking among full-time employees (42.7% for men versus 62.3% for women), figures that are inconsistent with the descriptives in Table 1.

Similarly, lines 247 to 250 compare the proportion of teleworkers by gender and the presence of children in the household. The proportions shown are surprisingly high: the percentage of men teleworking is around 40%, rising to 50% for women. The drastic reduction in sample size might explain these unusual results (e.g., “regarding the “children” variable, the proportion of teleworkers with and without children is very similar for men (42.4% and 39.8%, respectively) and women (56.1% and 50.0%, respectively)”).

10. Gender Gaps in Teleworking

Across the entire sample, the gender gap in teleworking adoption is relatively small (2.2 percentage points in favor of women). However, among full-time workers, the gender gap increases to 5 percentage points. Additionally, the probability of part-time work is 15 percentage points higher for women.

In households with children under 5, 42% of women work part-time compared to only 9% of men. In these households, gender differences in teleworking probabilities are minimal. However, as children age, the likelihood of part-time work decreases slightly among women (35% in households with children aged 6–11 and 26% in those with children aged 12–15). At these ages, the gender gap in teleworking rises to 6–7 percentage points. These findings raise doubts about the authors’ conclusion that work-life balance is not a significant factor in teleworking adoption.

11. Potential Additional Analyses

A possible avenue for exploration could involve focusing on the sample of full-time workers or combining the effects of part-time work (predominantly female) in the unequal development of teleworking. Descriptive tables are attached.

12. Policy Recommendations

The manuscript claims that teleworking has positive effects, such as reduced stress and increased flexibility. Where are the tables investigating this? Similarly, policy recommendations are made on issues not analyzed in the article.

13. Clarity and Presentation

The wording of some phrases is ambiguous:

a. Occupation appears as a factor influencing teleworking propensity (Asmussen et al., 2024). Why? Can the mechanisms making occupation relevant to teleworking be explained?

b. Similarly, education is noted as a factor to consider (Kley and Reimer, 2023). The reasons for its importance should be clarified, outlining how education affects teleworking.

Additionally, the presentation could be improved by numbering manuscript sections, completing tables with standard logistic regression statistics and sample sizes, and including titles and sources for the two final graphs.

Reviewer #2: I have carefully reviewed this manuscript and below is my decision.

-I consider that the paper's methodology is built on the basis of an appropriate theory of logistic regression. However, additional work could be done to improve the methodological arguments as follows:

- Despite the usefulness of the paper, it should undertake several diagnostic tests to examine basic logistic regression assumptions. In logistic regression, basic assumptions must be met, such as the independence of errors, the absence of multicollinearity and the lack of outliers. This can be easily achieved using any statistical or econometric software to enhance the quality of the paper. At present, the paper assumes all these assumptions are fulfilled. Although the findings can still be used where assumptions were violated, this could be explained to the readership.

- The other similar limitation of the paper is the assumption of linearity between the dependent variable and independent variables. This should be tested empirically and completed as a minor revision using diagnostic statistics to prove the linear relationship. Moreover, when the relationship is not linear, the regression findings should also be amended.

- The scale of the measurement should also be discussed more rigorously and clearly, such as which variables are on continuous scales and which ones are dichotomous, nominal, ordinal, interval or ratio. At the moment, the information for the readership on the data is limited.

-There is study that have examined logistic regression assumptions.

https://doi.org/10.5281/zenodo.8429022

- Are regression estimations weighted ones? If not, it is advisable to use sample weights (inflation factors) to make the results representative. Does the survey provide sample weights?

-A more detailed explanation should be written about the data set.

- All research has limitations. Prior to the ‘Conclusion’, please ensure to have a limitations section.

-The conclusion section should be written separately.

It can be published after corrections are made.

6. PLOS authors have the option to publish the peer review history of their article (what does this mean? ). If published, this will include your full peer review and any attached files.

**Do you want your identity to be public for this peer review?** For information about this choice, including consent withdrawal, please see our Privacy Policy .

Reviewer #1: No

Reviewer #2: No

---

## [Author Response · Author response to Decision Letter 1]

28 Feb 2025

Response to editor & reviewers (note: numbered lines in the author's comments to reviewers and in the editor's suggestions correspond to the original manuscript)

Editor

Suggestion

The manuscript's analytical approach is too descriptive. It must explain in more detail the causal relationships that affect telework adoption, such as between teleworking and the proposed affecting conditions.

Comment

This can be explained for several reasons. Firstly, teleworking tends to be more conditioned by structural factors of employment, such as sector of activity, educational level and type of contract, than by personal or family situation. In other words, the possibility of teleworking depends more on the characteristics of the job than on the fact of living alone or with other people.

Secondly, the flexibility of teleworking allows both singles and people in couples or with dependents to adapt to this modality without their cohabitation structure representing a significant barrier. Unlike other work dynamics that may be more influenced by household composition, telework appears to be similarly accessible to different types of households.

Likewise, these results could reflect a greater diversification of profiles within teleworking, in which different cohabitation groups manage to insert themselves without their personal situation clearly determining their participation in this type of work [line 257 and following]

This suggests that access to telework is not evenly distributed across all educational levels but is conditioned by educational background. It is possible that occupations that allow or facilitate this type of work require specialized skills, higher qualifications or belong to sectors where the use of digital technologies is essential. In this sense, telework could be reinforcing pre-existing inequalities in the labor market, favoring those with higher education and limiting access for those with lower educational levels [line 260 and following]

These results suggest that telework is not only linked to occupation, but also to job stability and workload. People with temporary contracts and reduced working hours, who tend to be in more precarious positions, have less access to this modality, which reinforces pre-existing labor inequalities. In addition, the fact that women have a greater presence in teleworking within these groups could be related to patterns of work-life balance and occupational segmentation, which raises questions about how teleworking may be reproducing gender dynamics in the labor market [line 271 and following]

Suggestion

There is a visible lack of figures and statistics related to the case of Spain. Therefore, a table showing the state of telework in Spain during the study's analysis period must be added.

Comment

The following amendments have been introduced (see line 167):

• A table showing Spain’s teleworking data from 2012 to 2021 has been added

• Next text has been added: In the case of Spain (see Table 2), there has been a feminization of regular telework since 2020, as in the countries of the second pattern described above. Prior to this, the proportion of men and women teleworking was comparable, although in most years there were slightly more men than women teleworking.It was subsequent to the confinement due to the pandemic that the gap between men and women widened significantly. While the pre-pandemic period exhibited no significant disparity between men and women, the year 2020 marked an increase of 2.2 percentage points, and one percentage point by the year 2021.

Therefore, the case of Spain exhibits a distinctive characteristic when compared to other countries within the same pattern. Consequently, the confinement and the pandemic resulted in a substantial increase in female teleworking, which was further fueled by the characteristics of the Spanish labor market, and gender variable has no major impact on the prevalence of telework (Duarte and Quirós, 2024; Eurofound, 2022a).

Suggestion

The proposed hypotheses must be formulated more explicitly to control for the proposed influential variables.

Comment

The following amendment has been introduced:

Text A has been replaced by Text B (see line 185):

Text A:

H.1.: Household composition acts as a moderator affecting the relationship between gender and propensity to telework, increasing in the case of women (Campdesuñer et al., 2023; Gohoungodji, 2022; Goldin, 2022; Ishino et al., 2022; Lyttelton et al., 2020; Pigini and Staffolani, 2019; Sun et al., 2023).

Text B:

H.1.: The composition of the household acts as a moderator in the relationship between gender and the propensity to telework, this influence being differentiated according to the presence or absence of children, as well as the form of cohabitation, whether living alone, with a partner or with more people. In households with children, the responsibilities of care and housework tend to fall on women, which increases their propensity to telework as a strategy to reconcile these demands (Campdesuñer et al., 2023; Lyttelton et al., 2020). On the other hand, in households without children, gender differences in the propensity to telework tend to be less pronounced, whereas in households with a couple or more people, assigned roles may amplify or reduce these differences (Goldin, 2022; Gohoungodji, 2023; Pigini and Staffolani, 2019; Ishino et al., 2021; Sun et al., 2023).

Text C has been replaced by Text D (see line 189):

Text C:

H.2.: Occupation acts as a moderating factor for gender differences in the propensity to telework (Asmussen et al., 2024; Chung and van der Lippe, 2020; Gálvez et al., 2021; Kley et al., 2023; López and Rodríguez, 2021; Lyttelton and Zang, 2020; Sostero et al., 2020; Lott et al., 2020).

Text D:

H.2.: Occupational category, defined by level of responsibility and skill, acts as a moderating factor in gender differences in the propensity to telework. Women in high-skill and higher-responsibility occupations are more likely to access telework due to the flexibility these positions often offer and the perception of trust involved in these roles (Kley y Reimer, 2023; Minkus et al., 2022; Chung and van der Lippe, 2020; Sostero et al., 2020). In contrast, in low-skilled occupations or occupations with lower levels of autonomy, gender differences in the propensity to telework are more marked, as these positions usually offer fewer options for work flexibility (Gálvez et al., 2021; López and Rodríguez, 2021).

Text E has been replaced by Text F (see line 193):

Text E:

H.3.: Education acts as a moderating factor in the propensity to telework (Kley and Reimer, 2023; Pigini and Staffolani, 2019; Santiago and Mergener, 2022).

Text F:

H.3.: Educational level acts as a moderating factor of the propensity to telework, differentiating access opportunities according to the degree of academic training. People with higher education are more likely to telework, given that the roles associated with this educational level tend to be less dependent on face-to-face work and more compatible with technological tools (Kley and Reimer, 2023; Pigini and Staffolani, 2019). In contrast, people with lower levels of education tend to occupy jobs that require manual skills or routine activities, limiting their ability to access teleworking

Suggestion

Although the logit model appears suitable for the paper's purposes, presenting marginal effects could offer a more intuitive result interpretation.

Comment

We have chosen to use a logistic regression model instead of presenting individual marginal effects for each variable, since our objective is to analyze the joint impact of the variables on the probability of teleworking. Marginal effects calculated in isolation may lead to misinterpretations, as they do not consider the interactions and relationships between the variables in the model.

To ensure a more accurate interpretation of the impact of each variable, we have presented the regression results in both tables and graphs, which allows us to visualize the actual effect of each predictor on the probability of teleworking, always controlling for the effect of the rest of the variables included in the model. This strategy provides a more rigorous and faithful view of the multivariate nature of the analyzed phenomenon.

Suggestion

Besides, additional diagnostic tests for logistic regression assumptions.

Comment

A logistic regression residual analysis was conducted to verify the validity of the logistic regression and the fulfillment of its assumptions. We have examined the behavior of the residuals using diagnostic plots, which show that the model is correctly specified and there are no systematic patterns suggesting fit problems.

The residual plots included in the analysis confirm that there are no indications of serious violations of the independence assumptions, absence of collinearity or poor model fit. In addition, the absence of influential outliers that could distort the estimates has been verified.

Thus, Graph 3 was included along with next text (see line 559):

Finally, an evaluation of the residuals of the logit model is conducted (Graph 3). It is observed that the residuals behave in accordance with the expected values, with no deviations identified, absence of collinearity or poor model fit.

Besides, in “Material and methods” section was included the next paragraph (see line 244):

Finally, an evaluation of the residuals of the logit model is conducted to verify the validity of the logistic regression and to determine whether the assumptions underlying the regression are fulfilled.

Suggestion

Discussion of measurement scales.

Comment

The measurement scales of different variables was pointed out (see lines 199, 208, 209, 216, 218, 224, 228 and 229)

Suggestion

The use of sample weights must be added.

Comment

The weights provided by the EWCTS survey are designed to ensure representativeness of the European population, correcting for possible biases derived from the sample design and non-response. However, given that our analysis focuses exclusively on Spain, the application of these weights may not be appropriate. The sampling structure and nonresponse patterns may differ within the Spanish subgroup, which would make the weights designed at the European level not correctly reflect the distribution of the Spanish population.

Furthermore, the use of these weights in a smaller sample such as ours carries a significant risk of inflating the type I error rate (false positives) by artificially reducing variances and making virtually all variables appear statistically significant. This is because the weights amplify the contribution of certain observations, affecting the estimation of standard errors in regression models.

According to methodological references, in studies whose main objective is to analyze relationships between variables, rather than to produce descriptive population estimates, the inclusion of weights is not always necessary and can even be detrimental if the weights have not been designed specifically for the subgroup analyzed. A valid alternative is to correctly specify the model by including the variables that affect the probability of response, which we have done by incorporating factors such as age, sex, region, occupation and sector in the analysis, as recommended by the EWCTS itself.

Sugestion

The reduction in the sample size from 2,903 to 1,320 individuals must be adequately explained.

Comment

The reduction of the sample size from 2,903 to 1,320 individuals is due to a refinement process based on consistency with the objectives of the study. In the original EWCTS survey, responses are distributed across multiple levels, and in some cases, the categorization is not clear or does not directly fit the variables in our model.

To avoid arbitrary subjective decisions and not artificially inflate the sample size, we have chosen to exclude those cases in which the responses could not be precisely assigned to the categories established in our analysis. This procedure guarantees the validity of the results and consistency in the classification of the variables, avoiding possible biases derived from an ambiguous assignment of responses.

The following amendment has been introduced, replacing text A by text B (see line 174):

Text A:

As a result of including the variables relevant to our analysis and selecting the data specific to Spain, we obtained a final sample size of 1,320 individuals.

Text B:

As a result of a refinement process based on consistency with the objectives of the study, the relevant variables and cases to our analysis were included, we obtained a final sample size of 1,320 individuals.

Suggestion

The empirical work needs to present results for the entire and gender-specific samples.

Comment

Following text has been introduced along with the new Table 4 regarding the entire sample analysis (see line 275):

The results of the logistic regression analysis for both men and women (Table 4) indicate the existence of high statistical significance of the education variable (“higher education level”: p<.001; z=7.959), and the labor variable such as “indefinite” (p<.001; z=5.073) related to “contract type”. Nevertheless, the other labor variable such as “full time” (p=.014; z=2.446) linked to “working time” has low statistical significance. The analysis also examined the association between other labor variable such as “occupation type” and telework, focusing on the occupation of scientific professionals, that have a high probability of telework for both men and women (0.91 ± 0.63 and 0.94 ± 0.74, respectively). All occupations under consideration have high Wald statistics and negative values, apart from “service and sales workers”, which do not demonstrate statistical significance.

Conversely, the relational variables such as having “children” (p=.55; z=0.599) and “living with a partner and more people” (p=.213; z=1.246) lack statistical significance in relation to telework.

Consequently, education and labor variables have been demonstrated to exert a substantial influence on telework patterns, indicating that workers with high level of education and indefinite contracts exhibit a greater propensity to telework, while those with full-time contracts demonstrate a slightly lower likelihood of doing so. On the other hand, although the labor variable related to occupation has a significant contribution to the predictive model of telework, it does so with an inverse character. This implies that the different occupations considered show a low probability to telework considering the occupation of scientific professionals, as a reference. Finally, the relational variables do not contribute to the predictive model related to telework.

Next text has been introduced along with the new Table 6 regarding the entire sample analysis (see line 307):

The table 6 presents the ratios and confidence intervals of the variables considered in the telework predictor model, considering both men and women samples. It indicates that the variables "education level-higher", "indefinite" related to the type of contract, and "full-time" referred to the type of working time have odds ratios greater than 1. This finding suggests that they directly influence the occurrence of teleworking. Besides, the strength of the relationship is more robust in the case of the "education level-higher" variable (OR=5.03) compared to the "indefinite" variable (OR=3.35) and the "full-time" variable (OR=1.76).

Conversely, the labor variables related to occupation have odds ratios of less than 1, so their influence on teleworking is inverse. Especially significant is the strength of the inverse relationship shown by the occupations “skilled agricultural, forestry and fishery workers”, “plant and machine operators, and assemblers”, “Elementary occupations”, and “armed forces occupations” with telework. This indicates that the probability of teleworking in these occupations is very low.

Finally, the test of multicollinearity, based on the Generalized Variance Inflation Factors (GVIF) indicates that there are no collinearity problems as their value is close to 1.

Suggestion

Some results in Tables 3 and 4 show striking gender differences that require further explanation.

Comment

Re

---

## [Editor Report · Decision Letter 1]

19 Mar 2025

PONE-D-24-47164R1Importance of occupation in the increase of gender differences in the advance of telework in SpainPLOS ONE

Dear Dr. Manzanera-Román,

Thank you for submitting your manuscript to PLOS ONE. After careful consideration, we feel that it has merit but does not fully meet PLOS ONE’s publication criteria as it currently stands. Therefore, we invite you to submit a revised version of the manuscript that addresses the points raised during the review process.

Although I acknowledge the authors' efforts to attend to the requirements posed by the reviewers and the editor, I consider that this version still needs additional reinforcements to satisfy the Journal's academic standards of quality and soundness to be published. Below are the comments expressed by the academic editor. Please submit your revised manuscript by May 03 2025 11:59PM. If you will need more time than this to complete your revisions, please reply to this message or contact the journal office at plosone@plos.org . Please include the following items when submitting your revised manuscript:

We look forward to receiving your revised manuscript.

Kind regards,

Humberto Merritt, PhD

Academic Editor

PLOS ONE

Journal Requirements:

Additional Editor Comments:

I refer to the first revised version of the manuscript titled "Importance of occupation in the increase of gender differences in the advance of telework in Spain," which analyses telework in Spain from a gender-based perspective and was submitted to its publication to the Journal PLOS ONE. Although I acknowledge the authors' efforts to attend to the requirements posed by the reviewers and the editor, I consider that this version still needs additional reinforcements to satisfy the Journal's academic standards of quality and soundness to be published. Hence, as the academic editor responsible for overseeing the submission process, I ask the authors to carry out a new revision to their manuscript to be published. Below are the comments expressed by the academic editor.

The transitions between sections, particularly from theory to methodology, are weak or insufficient. In this respect, the introduction should provide a more detailed discussion of the study's purpose and motives that drive the research question and hypotheses. Also, the paper should give more details on how missing data were handled. Then, a more detailed discussion on how the possible limitations of the dataset (e.g., self-reporting bias) would impact the methodology. Regarding the findings, the discussion section could better distinguish between expected and unexpected results to clarify contributions. Conclusions must also be enhanced because the manuscript offers valuable policy implications, particularly for Spain, where telework regulations are evolving. So, additional reflections regarding the research's policy implications for a broader context would be welcomed.

The manuscript's style and grammar must be improved. Several errors need corrections. For example, 1) the phrase "has triggered current digital shift" must be changed to "has triggered the current digital shift." 2) "earlier digital shifts, such as speed," instead of "earlier digital shifts, as speed." 3) "The Czech Republic," instead of "Czech Republic." 4) "telework possibilities," instead of "the possibilities of telework." 5) "... job characteristics seem to be factors to be considered...", instead of "...job characteristics seem to be a factor to be considered..." 6) "It will also depend on establishing boundaries between work and life," instead of "It will also depend on the establishment of boundaries between work and life." 7) "H.2.: The occupational category," instead of "H.2.: Occupational category." Etc.

On the other hand, be aware of the differences between American and British English in orthography spelling, for example, in "analyse and analyze" or "characterise and characterize." These cases highlight the need to be careful in the proper usage of English in the manuscript because wordy paragraphs that mix American English with British English weaken the comprehension and readability of the study. Regarding the references and citations, the formatting of some references needs standardization because of missing DOIs inconsistent style.

For these reasons, I judge that authors must remediate these newer observations to provide an improved version. In particular, they must attend to the following points:

Expand the introduction to provide an enhanced analysis of the study's purpose and motives that drive the research question and hypotheses

Add a brief explanation regarding how data limitations affect the results and how they were addressed

Improve readability by simplifying complex sentences.

Extensive proofreading must be carried out to address all grammar inconsistencies, including the mixed usage of American and British English.

Enhance policy discussion to highlight implications for labor regulations.

A paragraph dealing with policy recommendations must be added

Revise all citations to ensure no missing references in the text and the final section.

For you to instrument these changes in a timely fashion, the editor’s decision for the submitted manuscript is that it needs a minor revision of some parts of the paper. To this effect, you have 45 days to address these comments, requiring resubmission of the corrected version to Journal PLOS ONE.

---

## [Author Response · Author response to Decision Letter 2]

27 Mar 2025

Suggestion 1

Expand the introduction to provide an enhanced analysis of the study's purpose and motives that drive the research question and hypotheses

Comment 1

Taking into account the particularity of the Spanish case and the existing literature on telework from a gender perspective, this research adopts a cross-sectional approach. Using data from the European Working Conditions Survey, we seek to validate hypotheses on the moderating effect of the propensity to telework as a function of variables such as education, life stage (family structure and cohabitation) and work history (occupation, type of contract and working hours).

This study provides a novel perspective by analyzing how these factors influence the adoption of telework within the specific structure of the Spanish labor market. In doing so, it fills a gap in the literature by addressing insufficiently explored dimensions. In addition, its findings may contribute to the development of new theories on telework, as well as to the identification of areas that require further analysis. Finally, the results of this research can serve as a basis for the design of more effective policies that promote better working conditions, thereby fostering improved social welfare.

[line 215 and following]

Suggestions 2 & 3

- Improve readability by simplifying complex sentences.

- Extensive proofreading must be carried out to address all grammar inconsistencies, including the mixed usage of American and British English.

Comment 2 & 3

In accordance with the instructions provided, the manuscript has undergone a comprehensive revision to rectify all grammatical inconsistencies, including the utilisation of both American and British English.

It is asserted that this has resulted in a significant enhancement of the readability of the text.

Suggestion 4

Add a brief explanation regarding how data limitations affect the results and how they were addressed

Comment 4

This reduction was partly due to the limitations inherent in the EWCS dataset, such as the multi-level structure of responses, which in some cases did not align directly with the variables in the model. In order to address this issue and prevent systematic errors or biases resulting from ambiguous response assignment, cases that could not be accurately categorized were excluded. This approach was adopted to ensure the validity of the results obtained by avoiding the artificial inflation of the sample size.

In order to address the issue of missing data, a structured approach was adopted. Initially, an assessment was conducted to determine the extent and pattern of missing values in key variables. Cases with missing values in the dependent variable (teleworking status) were excluded, as they could not contribute to the analysis. For the independent variables, a complete-case analysis (listwise deletion) was applied, with observations retained only if complete data was available for all predictors. The selection of this method was driven by the necessity to preserve both the consistency and interpretability of the model, whilst ensuring that the final sample accurately reflected the population under study.

A sensitivity check was conducted to compare key characteristics (such as gender, occupation, and education level) between the full Spanish sample (2,903) and the final analytical sample (1,320). The distributions remained largely consistent, suggesting that the exclusion of incomplete cases did not introduce substantial bias.

[line 240 and following]

Suggestions 5 & 6

The results of this research can serve as a basis for the formulation of more equitable labor policies adapted to the needs of workers at different stages of their professional and personal lives. Although teleworking is regulated in detail in Spain as of Law 10/2021, of July 9, on telecommuting, studies such as this one can provide evidence for the revision of regulations on labor flexibility, work-life balance, and the protection of workers' rights in teleworking contexts. It is necessary to bear in mind that regulations must adapt to the changing reality of employment formulas, so research such as this can help to inform regulatory adaptations on the subject. They can also guide the design of measures to reduce gender inequalities in the access and use of telework, ensuring that this does not deepen pre-existing gaps in the labor market.

[line 674 and following]

Reference related to Ley 10/2021 has been included in References section:

Ley 10/2021, de 9 de julio, de trabajo a distancia, BOE, 164, Referencia: BOE-A-2021-11472

[line 873]

---

## [Editor Report · Decision Letter 2]

31 Mar 2025

Importance of occupation in the increase of gender differences in the advance of telework in Spain

PONE-D-24-47164R2

Dear Dr. Salvador Manzanera-Román,

We’re pleased to inform you that your manuscript has been judged scientifically suitable for publication and will be formally accepted for publication once it meets all outstanding technical requirements.

Kind regards,

Humberto Merritt, PhD

Academic Editor

PLOS ONE

Additional Editor Comments (optional):

After carefully revising the changes, corrections and adequations that the authors made to the two previously submitted versions of the manuscript titled "Importance of occupation in the increase of gender differences in the advance of telework in Spain," I confirm that this version has met the academic requirements and comments posed by the reviewers and the academic editor, and now fulfills the intellectual quality and originality criteria for publication in PLOS ONE.

So, please follow the submission instructions provided by PLOS ONE for further editorial directions.
---

## [Editor Report · Acceptance letter]

PONE-D-24-47164R2

PLOS ONE

Dear Dr. Manzanera-Román,

I'm pleased to inform you that your manuscript has been deemed suitable for publication in PLOS ONE. Congratulations! Your manuscript is now being handed over to our production team.

Kind regards,

on behalf of

Dr. Humberto Merritt

Academic Editor

PLOS ONE